# The Preventive Role of Glutamine Supplementation in Cardiac Surgery-Associated Kidney Injury from Experimental Research to Clinical Practice: A Narrative Review

**DOI:** 10.3390/medicina60050761

**Published:** 2024-05-03

**Authors:** Anca Drăgan, Adrian Ştefan Drăgan

**Affiliations:** 1Department of Cardiovascular Anaesthesiology and Intensive Care, Emergency Institute for Cardiovascular Diseases “Prof Dr C C Iliescu”, 258 Fundeni Road, 022328 Bucharest, Romania; 2Faculty of General Medicine, Carol Davila University of Medicine and Pharmacy, 8 Eroii Sanitari Blvd, 050474 Bucharest, Romania; dragan.adrian.stefan24@gmail.com

**Keywords:** kidney injury, inflammation, cardiorenal syndrome, cardiac surgery, multidisciplinary approach

## Abstract

Acute kidney injury represents a significant threat in cardiac surgery regarding complications and costs. Novel preventive approaches are needed, as the therapeutic modalities are still limited. As experimental studies have demonstrated, glutamine, a conditionally essential amino acid, might have a protective role in this setting. Moreover, the levels of glutamine after the cardiopulmonary bypass are significantly lower. In clinical practice, various trials have investigated the effects of glutamine supplementation on cardiac surgery with encouraging results. However, these studies are heterogeneous regarding the selection criteria, timing, dose, outcomes studied, and way of glutamine administration. This narrative review aims to present the potential role of glutamine in cardiac surgery-associated acute kidney injury prevention, starting from the experimental studies and guidelines to the clinical practice and future directions.

## 1. Introduction

EPIS-AKI, a recently prospective international observational multi-center clinical study, reported an acute kidney injury (AKI) incidence of 18.4% in major surgery (25.9% in cardiac surgery), with 33.8% persistent AKI and 8.7% patients requiring renal replacement therapy (RRT) [1]. Others found an AKI incidence of 20–30% in cardiac surgery [2,3]. Kidney Disease Improving Global Outcomes (KDIGOs) provided the AKI diagnostic and staging criteria in 2012 [4]. Although most of the cardiac surgery-associated AKI (CSA-AKI) cases are mild [5], identifying the high-risk patients is of utmost importance, because AKI has been independently associated with 30-day postoperative mortality [3,5]. The Randomized Evaluation of Normal versus Augmented Level Replacement Therapy (RENAL) study, which included cardiac surgery patients, revealed an overall mortality of 62.3% among all patients with AKI requiring RRT [2,6].

CSA-AKI has an impact on short and long-term complications as well. The patients with AKI had a longer intensive care unit (ICU) and hospital length of stay with higher hospitalization costs due to associated infections, prolonged mechanical ventilation, strokes, myocardial infarctions, and RRT [3,7]. Researchers reported the high costs associated with comorbidities and the progression of renal disease in the long run [3]. Even minor increases in creatinine levels after surgery (Δ creatinine 25–49% above baseline but <0.3 mg/dL) were found to lead to a two-fold increase in the risk of death and extended hospital stay [8]. Subclinical CSA-AKI was described as a new diagnostic entity. Novel diagnostic biomarkers might diagnose tubular damage without glomerular function loss [9,10]. These tools could assess early kidney damage before the increase in serum creatinine, allowing for a better assessment of the high-risk patients who could benefit from protective strategies. Haase et al. reported that 15–20% of patients without creatinine criteria for AKI still have acute tubular damage, associated with adverse outcomes [9]. In Marcello et al.’s study, 34.7% of patients were diagnosed with CSA-AKI by creatinine rise while 69.4% of patients with subclinical CSA-AKI using the plasmatic neutrophil gelatinase-associated lipocalin (NGAL) [11].

CSA-AKI can lead to chronic kidney disease (CKD) and also to end-stage renal disease. According to a study conducted by Xu et al., the CKD prevalence was significantly higher in CSA-AKI patients as compared to those who did not have AKI (6.8% vs. 0.2%, *p* < 0.001) in a two-year period [12]. AKI was still a risk factor for progressive CKD (RR 1.92, 95%, CI 1.37–2.69) even in patients with complete renal function recovery at discharge [12].

CSA-AKI prevention becomes crucial, especially when the treatment options are limited. Intravenous glutamine (GLN) administration was recently reported as an AKI prevention modality in high-risk cardiac surgery patients [13]. This approach is not novel, as the experimental studies previously demonstrated the myocardial and renal protective GLN effects. Moreover, GLN was considered a conditionally essential amino acid in stressful and inflammatory situations [14]. The cardiopulmonary bypass (CPB) used in cardiac surgery represents one of these situations, as Buter et al. have already demonstrated that the GLN levels decreased after CPB [15]. Although heterogeneous in their studied outcomes, patient selection criteria, timing, dose, and way of GLN administration, several clinical trials have studied GLN supplementation in cardiac surgery with promising results. This narrative review aims to present the potential role of GLN supplementation in CSA-AKI prevention, starting from the experimental studies and guidelines to the clinical practice and future directions.

## 2. Methods

This narrative review aims to explore the role of GLN in preventing CSA-AKI. It includes an overview of the pathophysiology of CSA-AKI, GLN generalities, and the specific guideline approaches, further presenting the experimental and clinical studies in this setting. This review is guided by the Scale for the Assessment of Narrative Review Articles (SANRA). We conducted a search on the online PubMed database using several keywords and phrases related to the use of glutamine in cardiac surgery and its effects on the heart and kidneys. These included: “glutamine in cardiac surgery”, “glutamine myocardial effects”, “glutamine renal protection”, “glutamine acute kidney injury”, “cardiac surgery-associated acute kidney injury”, “glutamine in intensive care unit”, “glutamine guidelines”, “glutamine ESPEN”, “glutamine ASPEN”, and “glutamine ERAS Cardiac Society”. The abstracts of the English-written articles were chosen based on their relevance to the subject matter, while case reports and case series were excluded. After eliminating duplicates, the selected full-length papers were further analyzed and evaluated for their relevance.

## 3. AKI in Cardiac Surgery: Insights into Pathophysiology

The complex pathogenesis of CSA-AKI is summarized in Figure 1.

Renal hypoperfusion that leads to renal hypoxia usually represents the first mechanism involved in CSA-AKI occurrence. The acute cardiovascular failure, met in cardiac surgery, could trigger cardiorenal syndrome type I [16]. The prolonged low cardiac output or hypotension might trigger AKI by activating the renin–angiotensin–aldosterone system, further leading to systemic vasoconstriction with more reduced renal blood flow [17,18]. The forced preoperative diuresis, excessive hemofiltration during CPB, blood loss, and vasoplegia can contribute to absolute or relative hypovolemia and consecutive impaired renal perfusion [2,18]. The hemodilution, hemolysis, rewarming, and the potential emboli related to CPB could participate in renal hypoperfusion [17]. Hypervolemia or right-side heart failure might also cause CSA-AKI, due to the intraoperative renal venous congestion [19,20].

The transient reduction in cardiac output, often met in cardiac surgery, is accompanied by I/R renal injury with the generation of reactive oxygen species (ROS), triggering inflammatory cascades and renal cell death. The I/R renal injury presents an ischemic phase, with renal tissue de-oxygenation and adenosine triphosphate (ATP) depletion, and a re-oxygenation phase that triggers ROS production, inflammatory cascade propagation, and renal tubular damage [21,22].

The iron overload due to hemolysis, the lipid peroxidation accumulation, and the inhibition of phospholipid hydroperoxidase glutathione peroxidase 4 (GPX4) triggers ROS that might further lead to renal ferroptosis, a nonapoptotic cell death that might lead to CSA-AKI [21].

The blood contact with an artificial surface, surgical trauma, I/R injury, blood loss, and transfusion represent factors that might trigger systemic inflammatory response syndrome (SIRS) by the complement system, cells, and endothelial activation, with cytokine production mediated by intracellular transcription factors (nuclear factor-κB) [23,24].

Although KDIGO criteria represent the current AKI diagnostic modality, several biomarkers were developed to allow for early assessment of CSA-AKI risk (NGAL [11,25], kidney injury molecule 1 (KIM-1) [26], dickkopf-3 (DKK3), liver fatty acid binding protein (L-FABP) [17], cystatin C (CysC) [25]), and PrevAKI studies reported that the urinary [TIMP-2]·[IGFBP7] > 0.3 defined the high-risk CSA-AKI [27,28]. The renal ultrasound point of care was also proposed to assess the CSA-AKI risk through evaluation of the arterial and venous renal profile [17,29,30,31,32,33,34,35,36].

## 4. Glutamine—Generalities

GLN, a l-α-amino acid containing five carbons, represents a non-essential amino acid. GLN concentration depends on the balance between its uptake, synthesis, and its tissue consumption. GLN availability might be impaired in hypercatabolic situations, as GLN represents an essential fuel for lymphocyte proliferation, cytokine production, macrophage phagocytic and secretory activities, and neutrophil bacterial killing. Endogenous GLN synthesis may be insufficient in major surgery, traumas, and sepsis. Therefore, GLN might be considered a conditionally essential amino acid as the demand might not match the supply. These aspects could lead to impaired immune function [14].

GLN is transported into cardiac myocytes via high-capacity transporters in the cardiomyocyte [37], where its catabolism was reported to be at least four times higher [38]. GLN is hydrolyzed into NH4+ and glutamate, converted further into α-ketoglutarate (α-KG) and succinate, providing the particular substrates for the Krebs cycle [26].

GLN represents the substrate for the arginine de novo production via the glutamine–citrulline–arginine pathway [39]. Arginine de novo production is crucial as it is the precursor of nitric oxide (NO) [39]. Several studies have explored the effects of intraoperative administration of NO on postoperative outcomes in cardiac surgery patients, including renal function improvement [2]. NO acts as a vasodilator, has endothelium protective effects, and also plays the role of mediator in host immune defense [39]. These effects are summarized in Figure 2.

GLN may be administrated orally/enterally (free GLN) or intravenously (GLN dipeptides). GLN dipeptides offer several advantages, such as stability during sterilization, prolonged storage, and a high range of solubility compared to free GLN [14]. Ligthart-Melis et al. demonstrated that the enteral administration of the GLN tracer resulted in a significantly higher intestinal fractional extraction of [15N] GLN, compared to its intravenously administration [40]. Moreover, the enteral administration of alanyl-[2-(15)N]glutamine contributes more to the de novo synthesis of arginine than intravenous infusion of the dipeptide does in humans [41].

Hasani et al. meta-analysis (2021) reported the GLN supplementation effects on the cardio-metabolic risk factors and inflammatory markers in adult and children populations with any health problem [42]. A more significant reduction in fasting plasma glucose and beneficial effect on IL-1 were related to the oral GLN route [42]. The intravenous GLN supplementation reduced hs-CRP more than oral administration [42], while no changes were reported in TNF-α and Il-6 levels [42].

In ICU, while hyperglutaminemia at admission was found to be an independent mortality predictor [43], the low plasma GLN level was reported to be associated with prolonged mechanical ventilation and nutritional support, and not with the length of stay (LOS) or mortality [44]. The enteral GLN supplementation did not ameliorate clinical outcomes in critical illness patients except for the reduction of hospital LOS in Liang and et al.’s meta-analysis [45]. On the other hand, parenteral glutamine dipeptide supplementation significantly reduced hospital mortality, infectious complication rates, and hospital LOS in Stehle et al.’s systematic evaluation [46].

However, the guidelines did not recommend routine GLN supplementation (Table 1). This is mainly the result of the REDOXS and MetaPlus studies. GLN was associated with an increase in mortality among critically ill patients with multiorgan failure [47] or mechanical ventilation [48].

## 5. Glutamine—Myocardial Protective Role

During cardiac surgery, a lot of myocardial changes take place at the molecular level. Suleiman et al. demonstrated that the intracellular GLN and ATP concentrations decreased, while the tissue lactate increased, irrespective of the cardioplegic solution [54]. Buter et al. reported that plasmatic GLN levels are significantly lower just after cardiac surgery compared to preoperative levels [15]. Later, Venturini et al. studied the amino acids myocardial concentration in biopsies collected from left (LV) and right (RV) ventricles before cardioplegic arrest and after reperfusion in patients undergoing mitral valve surgery [55]. Although the GLN baseline levels were higher in the dilated LV compared to the RV, they significantly decreased only in the LV in [55].

Wisschmeyer et al., studying the I/R myocardial effects in rats, found a significant lower ATP, glutamate, and reduced glutathione (GSH) levels in the myocardial tissue with lactate accumulation [56]. The cardiac output remained the same when GLN was administered 18 h before the I/R injury [56]. The authors suggested that the GLN mechanisms of action were represented by the conservation of the myocardial tissue metabolism, ATP, and reduced GSH levels, and by the stimulation of the heat shock protein (HSP) synthesis [56].

Bolotin highlighted the importance of timing, showing that 4 h of GLN pretreatment led to better cardiac output and coronary flow maintenance, which was superior when the pretreatment of 18 h was used [57]. In another study, adding GLN reduced I/R cell death with significantly increased cardiomyocyte HSP 72 expression without a decrease in intracellular oxidant generation [58]. Hayashi et al. demonstrated that preoperative GLN administration attenuated the CPB-induced inflammation by regulating NOS activity due to increased HSP70 expression [59]. Khogali et al. reported the recovery of the cardiac output as well in I/R injured heart rats treated with GLN [60], with a significant increase in the ATP/adenosine diphosphate (ADP) and the reduced GSH/oxidized GSH ratio by decreasing oxidized GSH [60]. Besides the oxidative mechanism, the researchers explained the beneficial effect of GLN through the maintenance of Krebs cycle activity due to conversion of glutamine to glutamate and subsequent formation of α-KG [60]. Watanabe et al. (2021) confirmed that in cardiomyocytes under oxidative stress, the glutaminolysis had cardioprotective effects by maintaining ATP and GSH levels by upregulation to compensate for the loss of α-KG and its replenishment in Krebs cycle [61].

Kristensen et al., on the contrary, reported GLN in their porcine model as an ineffective adjunctive therapy for severe myocardial ischemia and considered the consecutive significant increase in systemic resistance that followed GLN administration, as opposed to its myocardial protective effect [62]. The modulation of intracellular myocardial carbohydrate metabolism with an increasing de novo glycogen synthesis was a protective myocardial effect of GLN supplementation during reperfusion [63]. Liu et al. explained the GLN cardioprotective effect in an ex vivo rat perfused hearts by the increased flux through the hexosamine biosynthesis pathway (HBP), leading to increased O-linked N-acetyl-glucosamine (O-GlcNAc) levels [64]. GLN could also induce preconditioning for perioperative protection by enhancing the COX-2 activity [65] or by induction of HSP72 [66].

Li et al. (2015) found in their experimental study that a dramatic decline in intercellular GLN level and an increase in apoptosis occurred in diabetic I/R injured rat hearts [67]. GLN preinjury supplementation was associated with an intracellular ROS decrease, an increase in reduced GSH/oxidized GSH ratio in cytoplasm and mitochondria, and less apoptosis [67]. The inhibition of the transforming growth factor-β1-Smad3 pathway was also reported to be involved in cardiomyocytes GLN protection in a high glucose setting [68]. GLN could increase HSP 70 levels in the myocardium of rats with diabetes mellitus, as a protective mechanism, especially in the left heart chambers [69]. Lin et al. reported that the protective myocardial GLN effect was still working in an acidosis setting [70], while Drake et al. demonstrated the antiarrhythmic capacity of GLN supplementation in ischemic rabbit hearts [71].

Kou et al. (2016) found in their experimental study that I/R injury significantly increased miR-23a/b/c levels, while the glutamate, α-KG, and glutamate dehydrogenase (GDH) activity were significantly decreased [72]. In GLN-treated rats, the myocardial injury area was smaller compared with the control group. The researchers reported an inverse relationship between miR23 levels and the GLN metabolism, suggesting that anti-miR23 oligonucleotide might be a therapeutic agent against I/R injury in clinical practice as it restores the GLN metabolism [72]. Liu et al. (2017) found an overexpression of miR-200c with increased ROS levels and lower GLN metabolism following I/R heart injury, suggesting that blocking miR-200c might bring myocardial protective effects [73].

Cui et al. (2020) demonstrated that GLN protected myocardial cells previously exposed to I/R injury by activating the phosphatidylinositol 3-kinase/protein kinase B (PI3K/Akt) signaling pathway through increasing phosphorylated protein kinase B (p-AKT) and phosphorylated mammalian target of rapamycin (p-mTOR) levels [74]. GLN could enhance the cardiomyocytes’ proliferation ability by increasing the proliferating cell nuclear antigen (PCNA) level and reducing the P21 level [74]. The levels of inflammatory cytokines, tumor necrosis factor-alpha (TNF-α), and IL-6 were reduced and cell apoptosis was inhibited [74].

## 6. Glutamine—Renal Protection

Besides the cardiac involvement in renal function presented above, GLN supplementation was reported to be a direct renal protection especially in experimental studies. These are related to the activation of HSP-70 (heat shock protein 70) [75,76,77]. These effects, which include TNF-α, chemokines, and neutrophil infiltration, were eliminated with quercetin pretreatment, an HSP-70 inhibitor [75]. Only one GLN single dose could relieve renal I/R injury in rats by enhancing the HSP expression [76,77]. HSP-70 has immunosuppressive activity by downregulating nuclear factor-kappa B (NF-κB) activation [78]. Kim et al. found previously in an experimental study that GLN could ameliorate AKI tubular cell apoptosis by the c-Jun N-terminal kinase phosphorylation inhibition of 14-3-3 protein [79]. GLN prevented I/R renal injury by the induction of heme oxygenase-1(HO-1), an important antioxidant enzyme, via a p38MAPK-dependent pathway [80], and by downregulating miR-132-5p through cGMP-PKG signaling pathway [81].

GLN supplementation was protective in gentamicin AKI by increasing catalase, superoxide dismutase, glutathione peroxidase, and glutathione levels [82]. GLN administration 30 min before acetaminophen protected renal function [83].

Thomas et al. (2022) demonstrated that the renal GLN effects studied in a murine I/R AKI model were mediated by immunomodulation and not by increasing the renal blood flow, as the renal arterial resistivity index remained unmodified [84]. GLN was shown to act primarily on tubule epithelial cells rather than directly on neutrophils [84].

The reduction–oxidation capacity was affected, as well as the mitochondrial functionality, NAD metabolism, and apoptotic processes through transcriptomic and proteomic reprograming [84]. The gamma-glutamyl transferase 2 (Tgm2) and apoptosis signal-regulating kinase (Ask1) were reported as the main targets of apoptotic GLN signaling [84]. Chen et al. (2024) reported an increase in the expression of the ubiquinone oxidoreductase (complex I), with an NAD+/NADH ratio increase, improving the mitochondrial function by enhancing the oxidative phosphorylation process [85]. Yang et al. (2023) reported that the fluctuations in amino acid metabolism among metabolic pathways were observed in a renal I/R model [86]. GLN, converted to glutamate and further to α-KG, could replenish the tricarboxylic acid cycle. Thus, at the time of injury, the kidney might maintain the energy deficit by consuming the metabolic substrates [86].

The injury and remodeling of the cardiac and renal tissues depend on macrophage function. GLN can influence M1 macrophage polarization. Regarding M2 macrophages, GLN represents an energy source via its conversion to α-KG in the Krebs cycle and an epigenetic activator of M2-associated genes via histone demethylase Jmjd3 pathway [87]. Researchers reported T cell involvement in ischemic AKI occurrence and repair [88,89]. Targeting their metabolic reprograming by GLN might be a promising novel approach in AKI [90].

## 7. Glutamine in Cardiac Surgery—The Clinical Experience

Experimental studies have shown that administering GLN supplements has significant impacts, particularly in I/R cardiac and renal injury. GLN has been found to maintain myocardial tissue metabolism, ATP levels, and reduced GSH levels [56,60,61,82]. It also stimulates the synthesis of heat shock proteins [56,58,59,66] and replenishes α-KG in the Krebs cycle [8,61]. Reduced levels of inflammatory biomarkers and cell apoptosis were inhibited in experimental GLN studies [74]. The main mechanism behind GLN’s renal protection is immunomodulation [84]. Additionally, T cell reprograming [90] and macrophage polarization [87] were also reported as renal protective GLN mechanisms.

Several clinical trials have investigated the potential protective role of GLN in CSA-AKI by studying its anti-inflammatory effects, its ability to provide myocardial and renal protection, and its ability to improve glycemic control. However, these studies differ in terms of patient selection, endpoints studied, method of administration (intravenous or oral, single or in combination), dosage, and timing of GLN administration.

Engel et al. (2009) studied the GLN supplementation in open cardiac surgery patients that fulfilled one of the following criteria: age over 70 years, ejection fraction less than 40%, or mitral valve replacement [91,92,93]. The researchers used a high-dose L-alanyl-L-glutamine dipeptide (L-ALA-GLN) (0.5 g/kg/day GLN) after induction of anesthesia and continued it for three days. This action had only a minor influence on intracellular IL-2 and the expression of polarized intracellular T cell cytokine [93], and no effect on intracellular IL-1, IL-6, IL-8, and TNFα [92]. No other effects on C-reactive protein (CRP) level [92], mortality rate [93], postoperative infections number [93], vasopressor/inotropic support [92], ventilation time [92,93], renal function [93], SOFA score [92], and ICU length of stay [92] were found [93]. However, the high-dose GLN succeeded to maintain the GSH postoperative levels in contrast to the saline group [91]. In this regard, the researchers proposed a better selection of the patients who might benefit from GLN supplementation based on GLN deficiency detection [92].

The inflammatory aspects related to GLN supplementation in cardiac surgery were studied further by Efremon et al. The intestinal fatty acid binding protein (I-FABP), liver fatty acid binding protein (L-FABP), alpha glutathione S-transferase (αGST), HSP 70, the ventilation time, the hospital, and ICU length of stay did not differ in patients treated preoperatively with GLN 0.5 g/kg/day or with saline [94]. Svetikiene et al. reported more CD3+ T and CD4+ T cells when early enteral immunonutrition was used [95]. GLN was part of the immunonutrition complex (GLN 10 g, carbohydrate 10 g, b-carotene 1.7 mg, vitamin E 83 mg, vitamin C 250 mg, Zn 3.4 mg, Se 50 mg, and fiber 1.2 g) [95]. The T cell activation status was not affected [96]. Starting from the idea that nutritional deficiencies might induce inflammation and that cardiac surgery can bring a serum amino acid drop, Norouzi et al. (2022) studied the oral combination of GLN (7 g), β-hydroxy-β-methylbutyrate (HMB) (1.5 g), and arginine (ARG) (7 g) for 30 days before cardiac surgery [97,98]. A lower serum IL-6, erythrocyte sedimentation rate, hs CRP, and lymphocyte number were reported at the end of the study when compared to the placebo group [97]. No significant differences were reported regarding the number of neutrophils, lymphocytes, platelets, leucocytes, BUN, and creatinine levels [97]. Il-6 level was significantly lower in the intervention group, but TNF-α level was not [97]. SOFA score, the ICU, and hospital length of stay were lower in the GLN/HMB/ARG group [98].

Avoiding hyperglycemia is part of the KDIGO bundle preventive AKI actions [25]. GLN was studied in cardiac surgery from a glycemic control point of view. Hissa et al. (2011) reported that the intravenous administration of 250 mL L-ALA-GLN 20% in 750 mL saline over 3 h in the preoperative period improved glycemic control in patients with coronary artery occlusion, submitted to myocardial revascularization [99], while Lomivorotov et al. found no difference regarding insulin resistance, insulin sensitivity, and blood glycemia when a 0.4 g/kg/day of 20% L-ALA-GLN solution was used in diabetes mellitus II patients [100]. Recently, Ahmad et al. (2023) studied L-ALA-GLN (1.5 mL/kg body weight dose in 200 mL normal saline) in patients with uncontrolled diabetes presenting for urgent CABG [101]. Lower intraoperative (173.74 ± 19.97 mg/dL vs. 198.22 ± 14.64 mg/dL) and postoperative (162.36 ± 13.11 mg/dL vs. 176.13 ± 14.86 mg/dL) mean blood glucose levels, and lower mean total insulin requirements intraoperatively and postoperatively were reported in the intervention group [101].

Sufit et al. (2012) reported a significant decrease in troponin I (TnI) (at 24, 48, and 72 h), CK-MB (at 24 and 48 h), and myoglobin levels (at 24 h) postoperatively when patients were given 25 g of ALA/GLN orally twice a day for three days before surgery [102]. However, Lomivorotov et al. (2011) did not find this trend in TnI levels [100]. On the other hand, Chávez-Tostado et al. (2017) confirmed a significant reduction in postoperative myocardial damage and complications when GLN was used [103]. Fathi et al. (2018) also reported a significant postoperative reduction in TnI, CK, and CK-MB levels [104]. The authors also found a lower incidence of arrhythmic events, a decreased need for inotropes, an increased ejection fraction and blood pressure, and a shorter stay in the ICU [105]. Additionally, the time of GLN administration, preoperative or after anesthetic induction, did not significantly affect the results [104]. The myocardial protection qualities of intravenous GLN administration were also reported in low ejection fraction (31%–50%) patients undergoing elective on-pump CABG surgeries [26,105]. In the study by Parmana et al. (2022), the plasma lactate and TnI levels at 6 and 24 h post-CPB were significantly lower in the intervention group [105]. An examination of the tissue in the right atrial appendage revealed some important information. The levels of α-KG and anti-cardiac TnI were significantly higher, with a decreased myocardial injury score in the GLN group [105]. However, there was no significant change in the apoptotic index [105]. Furthermore, the cardiac index (CI) was significantly higher in the GLN group at 6 and 24 h after CPB [105]. Similarly, another study by Lomivorotov et al. reported a higher CI at 4 h after CPB, along with a lower systemic vascular resistance index, in patients who received GLN during cardiac surgery [106].

Although some studies did not find a significant effect on renal function [93,107], or in postoperative SOFA score [92], others reported other results [13,92,108]. Recently, Mostafa et al.’s (2023) study used intravenously GLN 0.4 g/kg/day for three days preoperatively and reported decreased postoperative NT-proBNP levels and hospital/ICU stays in mitral valve replacement surgery, without any significant implication upon postoperative human NGAL level or kidney function [107]. On the contrary, Weiss et al.’s (2023) randomized controlled, parallel-group, single-center, double-blind clinical study reported a significant decrease in markers related to kidney damage in cardiac surgery when GLN was used [13]. The high-risk patient selection was based on the urinary [TIMP2]*[IGFBP7] level measured at 4 h after CPB [13]. Although the overall AKI rate within 72 h was not different among groups, the urinary [TIMP-2]*[IGFBP7] was significantly lower in the GLN group compared to the control group (median, 0.18 [Q1, Q3; 0.09, 0.29], controls: median, 0.44 [Q1, Q3; 0.14, 0.79]; *p* = 0.01). [KIM-1] and [NGAL] were also significantly lower in the GLN group [13]. Table 2 summarizes data on GLN supplementation in cardiac surgery, related to CSA-AKI.

The experimental studies brought some facts about the importance of the GLN supplemental administration in CSA-AKI prevention. Although the heterogeneity of clinical studies is quite apparent in this setting, the Weiss et al. trial [13] reported encouraging results. GLN might be a solution in CSA-AKI prevention with an optimized section of patients. The urinary [TIMP2]*[IGFBP7] level ≥ 0.3 measured at 4 h after CPB represented the GLN supplementation criteria [13], as opposed to the GLN deficiency detection proposed by Engel et al. [92].

Based on recent experimental and clinical evidence, further studies are needed to investigate the potential of GLN in protecting against CSA-AKI. It is critical to carefully select patients and biomarkers, as routine GLN administration is not currently supported by guidelines.

Additionally, future research should explore the timing, method, and dosage of GLN administration adapted to the desired clinical result. In the meantime, we eagerly await the results of the studies conducted by Ogawa et al. [109] and Landoni et al. [110].

A study by Ogawa et al., conducted in an Asian cohort aged 65 years and above, focused on the use of an amino acid supplement (HMB 1200 mg, GLN 7000 mg, ARG 7000 mg) once or twice per day [109]. The dose varied based on the level of renal dysfunction and was administered for 14–28 days before an open-heart cardiac surgery [109]. The study aims to evaluate the supplement’s impact on the inflammatory and nutritional status, hospital mortality, ICU and hospital length of stay, and the incidence of postoperative complications, including renal function. The results of this study are still awaited [109].

The PROTECTION research represents the first multi-center randomized controlled study in this setting. This trial was designed to evaluate the relationship between amino acid use and kidney injury in cardiac surgery [110]. The primary outcome of the study was the incidence of AKI during hospital stays defined by KDIGO [110]. The treatment involved a continuous infusion of a balanced mixture of amino acids in a dose of 2 g/kg ideal body weight per day (up to a maximum of 100 g/day) from the time of admission to the operating room through to either death, start of RRT, ICU discharge, or 72 h after treatment initiation [110].

## 8. Conclusions and Future Directions

Supplementation of GLN can serve as a protective measure against cardiac surgery-associated AKI. Experimental studies have demonstrated its beneficial effects on both the myocardium and the kidneys, while clinical trials have reported positive results in reducing perioperative inflammation and oxidative stress, maintaining myocardial function, and optimizing glycemic control during cardiac surgery. Further studies are required to define patient selection criteria, dosage, timing, and route of administration, as routine GLN supplementation is currently not supported by guidelines.

## Figures and Tables

**Figure 1 medicina-60-00761-f001:**
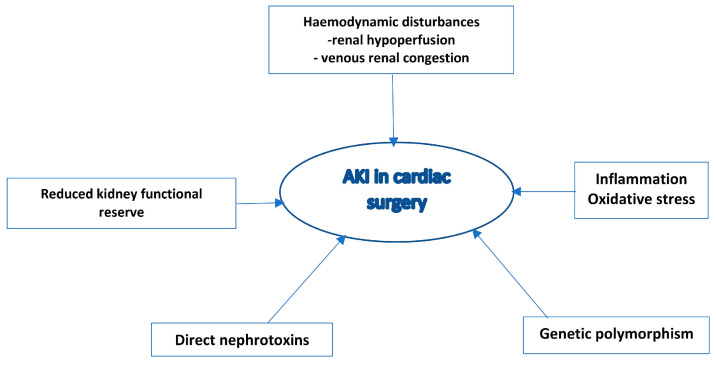
Pathophysiology of acute kidney injury associated with cardiac surgery. Abbreviations: AKI, acute kidney injury.

**Figure 2 medicina-60-00761-f002:**
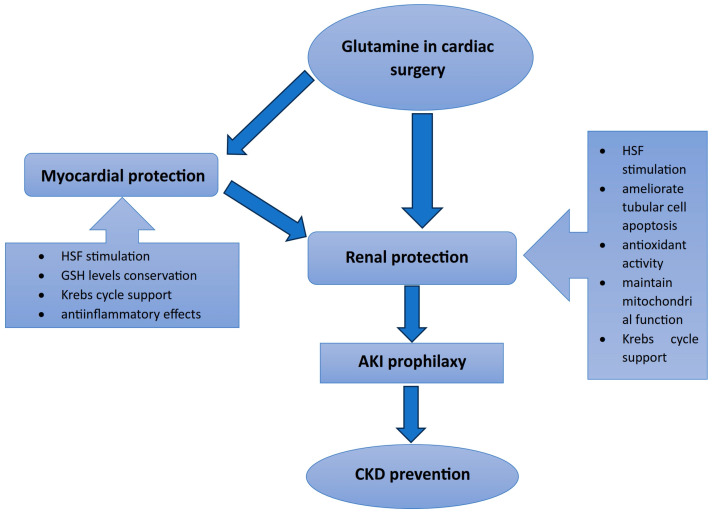
The main GLN effects involved in CSA-AKI protection. The main protective renal mechanisms of glutamine in cardiac surgery. Abbreviations: AKI, acute kidney injury; CKD, chronic kidney disease; GSH, glutathione; HSF, heat shock protein.

**Table 1 medicina-60-00761-t001:** The relevant guideline recommendations regarding GLN supplementation in the cardiac surgery, AKI, and ICU setting.

Guideline	Year	Recommendation	Grade of Recommendation
ESPEN guideline on clinical nutrition in hospitalized patients with acute or chronic kidney disease [49].	2021	In critically ill patients with AKI, AKI on CKD, additional high dose parenteral GLN shall not be administered [49].	Grade A^−^
ESPEN practical guideline: Clinical nutrition in surgery [50].	2021	Parenteral GLN supplementation may be considered in patients who cannot be fed adequately enterally and require exclusive PN [50]. Currently, no clear recommendation can be given regarding the supplementation of oral GLN (0) [50].	Grade 0
ESPEN practical and partially revised guideline: Clinical nutrition in the intensive care unit [51].	2023	In critically ill trauma patients, additional EN doses of GLN (0.2–0.3 g/kg/d) can be administered for the first five days with EN [51].	Grade 0
		In ICU patients except burn and trauma patients, additional enteral GLN should not be administered [51].	Grade B
		In unstable and complex ICU patients, particularly in those suffering from liver and renal failure, parenteral GLN-dipeptide shall not be administered [51].	Grade A
		In patients with burns >20% body surface area, additional GLN enteral (0.3–0.5 g/kg/d) should be administered for 10–15 days as soon as EN is commenced [51].	Grade B
Adult Cardiac Surgery-Associated Acute Kidney Injury: Joint Consensus Report (POQI and ERAS Cardiac Society) [5].	2023	No recommendations on GLN supplementation.	
SCCM and ASPEN Guidelines for the Provision and Assessment of Nutrition Support Therapy in the Adult Critically Ill Patient [52].	2016	The immune-modulating enteral formulations (arginine with other agents, including GLN) should not be used routinely in the medical ICU. Consideration for these formulations should be reserved for TBI and perioperative patients [52].	Quality of Evidence: Very Low
		The guideline suggested that ICU patients with AKI be placed on a standard enteral formulation [52].	Expert consensus
ASPEN Guidelines for the provision of nutrition support therapy in the adult critically ill patient [53].	2022	No relevant updates on GLN supplementation.	

Abbreviations: AKI, acute kidney injury; ASPEN, American Society for Parenteral and Enteral Nutrition; CKD, chronic kidney disease; EN, enteral nutrition; ERAS, enhanced recovery after surgery; ESPEN, European Society for Clinical Nutrition and Metabolism; GLN, glutamine; ICU, intensive care unit; POQI, perioperative quality initiative; PN, parenteral nutrition; SCCM, Society of Critical Care Medicine; TBI, trauma brain injury.

**Table 2 medicina-60-00761-t002:** The relevant clinical trials studying GLN supplementation in cardiac surgery, related to CSA-AKI.

Authors (Year)	Ref.	GLN	Studied Population		Main Findings
			No.	Type of Surgery	
Engel et al. (2009)	[91]	iv	60	⮚ Elective cardiac surgery	⮚ Maintained the GSH levels postoperatively
Engel et al. (2009)	[92]	iv	60	⮚ Elective cardiac surgery	⮚ No effects on CRP, SOFA score, circulation support, postoperative ventilation time, and ICU length of stay
Engel et al. (2009)	[93]	iv	78	⮚ Elective cardiac surgery	⮚ No effects on postoperative infections, mortality rate, ventilation time, or renal function.
Lomivorotov et al. (2011)	[106]	iv	50	⮚ Elective on pump CABG	⮚ Cardioprotective properties
Hissa et al. (2011)	[87]	iv	22	⮚ Elective on-pump CABG	⮚ Improvement of glycemic control
Sufit et al. (2012)	[102]	po	14	⮚ Elective cardiac surgery	⮚ Reduced myocardial injury ⮚ Reduced clinical complications
Lomivorotov et al. (2013)	[100]	iv	64	⮚ Elective on-pump CABG	⮚ No cardioprotective properties ⮚ No better glycemic control
Efremov et al. (2014)	[94]	iv	24	⮚ Elective CABG	⮚ No protection on gastrointestinal tract ⮚ Unchanged I-FABP, L-FABP, αGST, HSP 70
Chávez-Tostado et al. (2017)	[103]	po	28	⮚ Elective cardiac surgery	⮚ Less postsurgical complications, ⮚ Myocardial protection
Svetikiene et al. (2021)	[95]	po	55	⮚ Elective cardiac surgery	⮚ Higher counts CD3+ and CD4+ T cells
Svetikienė et al. (2021)	[96]	po	55	⮚ Elective cardiac surgery	⮚ TNF-α, IL-6, IL-10, PCT, and CRP, CD69+ not significantly changed.
Parmana et al. (2022)	[105]	iv	60	⮚ Elective on-pump CABG	⮚ Myocardial protection
Norouzi et al. (2022)	[98]	po	60	⮚ Elective cardiac surgery	⮚ Enhanced recovery, ⮚ Reduced myocardial injury, ⮚ Decreased the in-hospital and ICU stay
Norouzi et al. (2022)	[97]	po	60	⮚ Elective cardiac surgery	⮚ Attenuated the TNF-α, IL-6, hs-CRP, ESR increase ⮚ Significant reduction of IL-1. ⮚ No BUN or creatinine change.
Parmana et al. (2023)	[26]	iv	60	⮚ Elective on-pump CABG	⮚ Myocardial protection
Ahmad et al. (2023)	[101]	iv	93	⮚ Urgent CABG	⮚ Improvement of glycemic control
Weiss et al. (2023)	[13]	iv	64	⮚ Elective on-pump cardiac surgery	⮚ Decreased renal stress ⮚ Decreased damage renal biomarkers in high-risk AKI patients

Abbreviations: AKI, acute kidney injury; BUN, blood urea nitrogen; CABG coronary artery bypass graft surgery; CRP, C reactive protein; ESR, erythrocyte sedimentation rate; GLN, l-glutamine; GSH, glutathione; hs, high sensitive; HSP 70, heat shock 70; I-FABP, intestinal fatty acid binding protein; ICU, intensive care unit; IL-1, interleukin-1; IL-6, interleukin-6; IL-10, interleukin-10; iv, intravenously; L-FABP, liver fatty acid binding protein; No., number; PCT, procalcitonine; po, orally; Ref., reference; TNF-α, tumor necrosis factor-alpha; TnI, troponin I; SOFA, sequential organ failure assessment; αGST, alpha glutathione S-transferase.

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
