# Peer review of "The Preventive Role of Glutamine Supplementation in Cardiac Surgery-Associated Kidney Injury from Experimental Research to Clinical Practice: A Narrative Review"

_medicina, 2024, doi:10.3390/medicina60050761_

Round 1

Reviewer 1 Report

Comments and Suggestions for Authors

The authors present a review of “The preventive role of glutamine supplementation in cardiac surgery-associated kidney injury from the experimental research to the clinical practice”. The feedback provided by this reviewer regarding this work is delineated below.

The abstract necessitates a more detailed revision as it currently lacks coherence.

It is suggested to include your search strategy in the manuscript and update your search not to miss any articles.

The article would benefit from a thorough revision to enhance the quality of English, grammar, and verb tense consistency.

Comments on the Quality of English Language

 Moderate editing of the English language required

Reviewer 2 Report

Comments and Suggestions for Authors

It is very interting article.

Reviewer 3 Report

Comments and Suggestions for Authors

The main doubt that arises when revising this review is methodological: the authors point out that it is a narrative review, but the information they present is exhaustive. Have the authors consulted the SANRA Guidelines for narrative reviews? If you have used it, please state in the Introduction or Methods section that you have followed SANRA guidance. For example: 'This review is guided by the Scale for the Assessment of narrative review articles (SANRA) https://doi.org/10.1186/s41073-019-0064-8.

Some acronyms should be reviewed: KNIGO (line 32), or CAS_AKIA (line 45)

AKI in cardiac surgery: insights into pathophysiology: Reading this section is complicated and seems to be a development of unconnected ideas. I think it could be summarized and ordered.

Glutamine-Generalities: All information on glutamine metabolization is outside the scope of this review. From my point of view, the authors should focus this section on the importance of glutamine in critically ill patients, so they should summarize this section. Figure 1 is appropriate. I find the information provided by table 1 interesting.

Once the two previous sections have been summarized, the potential reader will be able to focus on sections 4, 5 and 6. From my point of view, these sections are the fundamental basis of this article.

Duplication of the information provided in Table 2 with that of the text should be avoided. It should remembered that tables usually provide information in a more concise and useful way for the reader.

The information provided in the conclusions is practically superimposable to what is known before the review. I recommend reinforcing the conclusions based on what has been reviewed.

Author Response

Thank you for your observations! We revised the manuscript, including references. The changes are highlighted in the revised manuscript.

According to your suggestion, we made the requested changes. 

We added a new section called Methods after the Introduction to explain our search strategy, including the adherence to the SANRA guidance. We have also added to the title ": a narrative review" at the suggestion of the other reviewer. 

The acronyms have been reviewed.

The new Sections 3 and 4 (former Sections 2 and 3) were summarized. This will help the reader to focus on the following sections easily. Additionally, a new figure has been added in Section 3 to better explain the complex pathophysiology of AKI in cardiac surgery. The information on glutamine metabolization was minimized and new ideas were added regarding the importance of glutamine in the ICU setting.

Table 2 has been restructured to provide more concise and useful information for the reader.

We have reformulated the Conclusion to make it more clear and concise.

Dear Reviewer, we hope that the current version of the manuscript appropriately addresses your suggestions and that now you will find our paper suitable for publication in Medicina.

Thank you for kindly considering our manuscript!

Looking forward to your decision!

Reviewer 4 Report

Comments and Suggestions for Authors

The manuscript is well written. Suggested is to add to the title;  ": a narrative review". 

L 51: delete "is also significant as it" and "even"and add "also to" giving: CSA-AKI can lead to chronic kidney disease and also to end-stage renal disease.

par 6: the focus is very much on inflammation, but not on protection of AKI per se. Advise is to condense the text with a special focus on the clinical experience in relation to CSA-AKI with the role of Glutamine suppletion.

L 393 etc.: The bottom-line is not well supported by the  literature per se, but is more the natural result based on the focus of this review. Why should future trials examining the effects of glutamine  suppletion focus on renal function per se? It would be interseted and strengthen the conclusion to have the advise of this reviewers on which dose, which timing and which biomarkers should be selected for such studies based on this review.

Comments on the Quality of English Language

The manuscript is well written

Round 2

Reviewer 3 Report

Comments and Suggestions for Authors

I congratulate the authors. The manuscript has improved considerabily after the changes made. All the issues arisen in the previous revision have been addressed.